# Electrical Properties of Two Types of Membrane Component Used in Taste Sensors

**DOI:** 10.3390/s21248343

**Published:** 2021-12-14

**Authors:** Zhanyi Xiang, Yifei Jing, Hidekazu Ikezaki, Kiyoshi Toko

**Affiliations:** 1Graduate School of Information Science and Electrical Engineering, Kyushu University, 744 Motooka, Nishi-ku, Fukuoka 819-0395, Japan; jing.yifei.623@s.kyushu-u.ac.jp; 2Intelligent Sensor Technology, Inc., 5-1-1 Onna, Atsugi-shi 243-0032, Japan; ikezaki.hidekazu@insent.co.jp; 3Institute for Advanced Study, Kyushu University, 744 Motooka, Nishi-ku, Fukuoka 819-0395, Japan; toko@ed.kyushu-u.ac.jp; 4Research and Development Center for Five-Sense Devices, Kyushu University, 744 Motooka, Nishi-ku, Fukuoka 819-0395, Japan

**Keywords:** taste sensor, lipids, dissociation of H^+^, cation selectivity, membrane resistance, Hofmeister series

## Abstract

The lipid phosphoric acid di-n-decyl ester (PADE) has played an important role in the development of taste sensors. As previously reported, however, the concentration of PADE and pH of the solution affected the dissociation of $H^{+}$, which made the measurement results less accurate and stable. In addition, PADE caused deterioration in the response to bitterness because PADE created the acidic environment in the membrane. To solve these problems, our past study tried to replace the PADE with a completely dissociated substance called tetrakis [3,5-bis (trifluoromethyl) phenyl] borate sodium salt dehydrate (TFPB) as lipid. To find out whether the two substances can be effectively replaced, it is necessary to perform an in-depth study on the properties of the two membranes themselves. In this study, we fabricated two types of membrane electrodes, based on PADE or TFPB, respectively, using 2-nitrophenyl octyl ether (NPOE) as a plasticizer. We measured the selectivity to cations such as ${\mathrm{Cs}}^{+}$,${\;K}^{+}$,${\;\mathrm{Na}}^{+}$ and ${\mathrm{Li}}^{+}$, and also the membrane impedance of the membranes comprising PADE or TFPB of the different concentrations. As a result, we found that any concentration of PADE membranes always had low ion selectivity, while the ion selectivity of TFPB membranes was concentration-dependent, showing increasing ion selectivity with the TFPB concentrations. The ion selectivity order was ${\mathrm{Cs}}^{+}>K^{+}>{\mathrm{Na}}^{+}>{\mathrm{Li}}^{+}$. The hydration of ions was considered to participate in this phenomenon. In addition, the membrane impedance decreased with increasing PADE and TFPB concentrations, while the magnitudes differed, implying that there is a difference in the dissociation of the two substances. The obtained results will contribute to the development of novel receptive membranes of taste sensors.

## 1. Introduction

In the 1960s to 1990s, with the continuous improvement of membrane theory, many ion-selective electrodes were invented, and are still used to distinguish different ions today. Some of them were of the liquid-membrane type, based on lipid polymer membranes [1,2,3,4,5,6]. However, there are also lots of restrictions on the conditions, such as temperature and pH [7,8,9].

Toko et al. found that lipid polymer membranes could behave in the same way as human gustatory cells, where the membrane potential can be used to extract the information of tastes [10,11,12,13,14]. They successfully developed a taste sensor equipped with lipid polymer membranes in 1989 [13]. Different from the ion-selective sensors, the taste sensor can distinguish and quantify five basic tastes, such as sourness, sweetness, saltiness, bitterness, and umami. Each basic taste is measured by one kind of taste sensor where the membrane is specifically constructed to measure the taste intensity based on a large quantity of possible substances producing tastes [10,11,12,15,16].

Not only can this taste sensor be used under most conditions, but it can be reused many times without affecting the measurement accuracy [17]. However, a problem has appeared that must be overcome in taste sensors using the membrane comprising phosphoric acid di-n-decyl ester (PADE) as the artificial lipid. Because of the $H^{+}$ dissociation of the phosphate group, PADE is responsible for the negative charges at the surface of the membrane [18]. This lipid has been used in our taste sensors for more than 20 years [19]. The degree of dissociation will be influenced by the lipid concentration of the membrane and the pH of the solution because it is a weak electrolyte. Therefore, the accuracy and durability of sensors using such kinds of lipids can become unstable. In recent studies, Wu et al. found that adding too much PADE can create an acidic environment, which can lead to hydrolysis of the plasticizer, resulting in deterioration of the response to bitter substances [10]. Nakatani et al. also found that due to the partial dissociation of PADE, the measurements of amphoteric electrolyte samples, such as amino acids, can cause changes in $H^{+}$ dissociation of PADE, resulting in unexpected changes in membrane potential, which affects the response [20]. They tried to improve this unfavorable response by adjusting the lipid concentration and plasticizer type, but none of these methods solved the problem satisfactorily.

To find a way to improve our taste sensors, a new substance is needed to replace PADE, and hence tetrakis [3,5-bis(trifluoromethyl)phenyl] borate sodium salt dehydrate (TFPB) was selected as a candidate [21]. TFPB is featured with little nucleophilicity [22], and high chemical durability because of its fluoroaromatic borate structure [23]. It is stable under acidic environments [24]. It is often used as an anion phase-transfer catalyst or as an anion excluder in ISEs [25]. As a completely dissociative anionic lipophilic additive, it is a good choice for cation sensing [26]. Its completely dissociative properties make it possible for it to stay unchanged under acidic conditions. In fact, the membrane using TFPB presented long-term stability, and also showed high selectivity and sensitivity for pharmaceutical bitterness [21]. As a result, the chemical durability and completely dissociative property of TFPB can be considered to make it a candidate to solve the problems concerned on PADE, which are used in taste sensors for sourness, bitterness, and umami [11,12].

Before TFPB is utilized in lipid polymer membranes developed for a commercialized taste sensor, it is necessary to understand the changes in basic properties that occur when PADE is replaced with TFPB. Here, we aim to find and compare the electrical properties of a partially dissociated membrane using PADE and a completely dissociated one using TFPB. For this purpose, we measured the responses to alkali metal ions with the aid of the taste sensing machine TS-5000Z, and obtained the ion selectivity coefficients, and also measured the membrane impedance of the two membranes. As a result, we found that the PADE membranes showed low ion selectivity, irrespective of the contained PADE concentration of the membrane, while the TFPB membranes showed increasing ion selectivity with increasing TFPB concentrations. This order was ${\mathrm{Cs}}^{+}>K^{+}>{\mathrm{Na}}^{+}>{\mathrm{Li}}^{+}$, in accordance with the well-known Hofmeister series. The membrane impedance slightly decreased with increasing PADE, while it largely decreased with TFPB concentrations. The fact that PADE membranes with impedances as high as $10^{6}\Omega \cdot {\mathrm{cm}}^{2}$ have low ion selectivity at any PADE concentration is in line with the TFPB membrane showing high impedance at lower TFPB concentrations. These findings will contribute to the development of a novel taste sensor robust against changes in the circumstance in lipid polymer membranes.

## 2. Materials and Methods

### 2.1. Reagents

Phosphoric acid di-n-decyl ester (PADE) was purchased from Tokyo Chemical Industry Co., Ltd. (Tokyo, Japan). Tetrakis [3,5-bis(trifluoromethyl)phenyl] borate sodium salt dehydrate (TFPB) and polyvinyl chloride (PVC) were purchased from FUJIFILM Wako Pure Chemical Corporation (Osaka, Japan). 2-Nitrophenyl octyl ether (NPOE) and tetrahydrofuran (THF), which were used as the supporting material and the organic solvent, were purchased from Sigma-Aldrich Japan G.K. (Tokyo, Japan. Figure 1 shows the structures of these substances used to make up the membrane.

Measurement samples are as follows. Potassium chloride (KCl) and sodium chloride (NaCl) were purchased from Kanto Chemical Co., Inc. (Tokyo Japan). Lithium chloride (LiCl) and cesium chloride (CsCl) were purchased from FUJIFILM Wako Pure Chemical Corporation. All samples to be tested were dissolved in pure water. The details of sample solutions will be recorded in Section 2.4.

### 2.2. Lipid Polymer Membrane

In this study, two types of lipid polymer membranes were fabricated; i.e., one using a partially dissociated lipids PADE, called PADE membrane, and the other using a completely dissociated lipids TFPB, called TFPB membrane. Both the PADE and TFPB membranes contained 400 μL (3 M) NPOE as a plasticizer, and 200 mg PVC as a polymer supporting reagent. To investigate the effect of lipid concentration on the electrical properties of membranes, the PADE membranes were fabricated with 10 different concentrations from 0.03 mM to 300 mM, and the TFPB membranes from 0.01 mM to 300 mM. A blank membrane based on NPOE without lipid was made and measured as a reference. The PADE and TFPB concentrations are shown in Table 1. The reason for the different concentration ranges of the two membranes is because the ion selectivity of PADE membranes always varied less, and hence the lower concentration experiments were not necessary, while the ion selectivity of TFPB membranes varied more and low concentration experiments were needed to confirm the trend. The experimental results about this point will be detailed in Section 3.3 and Section 3.4.

The fabrication steps were as follows. Firstly, lipids, NPOE, and PVC were dissolved in the solvent THF and stirred for one hour. Secondly, the solution was poured into a 45 mm Petri dish and dried for three days in a draft chamber to let the THF be volatilized. Finally, the membranes were cut into pieces that could be pasted onto a sensor probe. An adhesive made by mixing 10 mL THF and 800 mg PVC was used to attach the membranes to the probes. After attaching the membrane to the probe, it was necessary to wait for 24 h to make it completely attached. From one Petri dish, we made four sensor electrodes to measure the sample. The thickness of the lipid polymer membranes was about 0.35 mm.

### 2.3. Measurement Procedure of Taste Sensor

In this study, a commercialized taste sensing system called TS-5000Z (Intelligent Sensor Technology, Inc., Kanagawa, Japan) was used for the measurement. This measuring instrument uses the two-electrode method, which has at most 4 sensor electrodes and a reference electrode on the same measurement cell, to measure the membrane potential differences between sensor electrodes and reference electrode as output. The construction of the electrochemical cell is Ag-AgCl| 3.33 M KCl + satd. AgCl|membrane|sample solution|3.33 M KCl + satd. AgCl|AgCl-Ag, and is shown in Figure 2.

The lipid polymer membranes were pasted onto a hollow sensor probe. The actual measured area of lipid polymer membranes is about 0.4 ${\mathrm{cm}}^{2}$. Both the sensor electrodes and the reference electrodes were injected with 200 μL 3.33 M KCl and saturated AgCl.

The images of the taste sensor system TS-5000Z and the measurement procedure are shown in Figure 3 and Figure 4, respectively. Before the measurement, we had a preconditioning process to re-arrange the lipids in the inside of the membrane surface, so that a stable measurement potential could be obtained. In this process, the sensor electrodes and reference electrodes were immersed in standard solution (30 mM KCl and 0.3 mM tartaric acid) for 72 h. In the measurement section, step 1, the measurement cell was immersed in standard solution (30 mM KCl and 0.3 mM tartaric acid) for 30 s to obtain a membrane potential called *V*_r_. In step 2, the measurement cell was immersed in sample solution for 30 s to obtain a new membrane potential called V_s_. The difference between *V*_s_ and *V*_r_ (*V*_s_ − *V*_r_) is defined as the relative value. In this study, the relative value was used to obtain the response value. In step 3, to restore the electrode surface to its pre-measurement condition, the measurement cell was washed with a cleaning solution (100 mM HCl and 30 vol% ethanol) [17]. Steps 1 to 3 were repeated until all the samples were measured, and it is defined as one measurement cycle. In this study, one measurement cell contained 4 sensor electrodes and 1 reference electrode, and five measurement cycles were needed for one measurement procedure. The average of the third to fifth measurement cycle was calculated as the response value. The standard deviations were calculated from n=4 (sensor electrodes)×3 (cycles)=12 values in the same way as previous studies [10,17,20,27].

### 2.4. Measurement of the Selectivity Coefficient

To investigate the electrical properties of two membranes, responses to different cations were measured and the ion selectivity was calculated. The sample concentrations are shown in Table 2. All sample solutions were prepared with chloride anions and were dissolved in pure water, which means that the pH of the sample solutions was about 5.2. The selectivity coefficients were calculated using the separate solution method (SSM) [28], where the response values from 10 mM to 1000 mM were utilized and those of potassium chloride were adopted as the reference.

### 2.5. Measurement of Membrane Impedance

In the measurement procedure of membrane impedance, Autolab PGSTAT302F was used to provide the measuring environment, and the user interface NOVA was used to automatize and analyze the measuring procedure. The details of settings are listed in Table 3. The impedance of 0.1 Hz was used as the dedicating figure of membrane impedance.

## 3. Results and Discussion

### 3.1. Sensor Response to Different Alkali Metals in the PADE Membrane

Figure 5a shows the sensor response of the blank membrane containing no lipid to different cations. Figure 5b,c show the sensor responses of membrane electrodes with low and high PADE concentrations to different cations, respectively.

Figure 5a,b show that the membrane electrodes with low PADE concentration (0.03 mM) and the blank membrane have almost the same response; this implies that the electrical properties of the membrane with low PADE concentration are almost the same as those of the blank membrane, which depends on the plasticizer NPOE. The sensor responses of both blank membrane (a) and low-PADE membrane (b) electrodes increased logarithmically with an increasing concentration of each alkali metal salt. It was also found that there is a large response value for CsCl. As other research on PVC membranes have also found cation permselectivity, it is reasonable to speculate that the rise in response is due to charged impurities contained in the membrane components such as NPOE [29,30,31]. In addition, Figure 5c shows that the sensor with high concentration of PADE membrane (30 mM) has a higher response to alkali metals than both the low-PADE and blank membranes. The reference potentials (*V*_r_) of the measurements are shown in Table 4 and can be used to calculate the response value (*V*_s_ − *V*_r_).

The KCl response (*V*_s_ − *V*_r_) of the high-PADE membrane electrodes (Figure 5c) is about −60 mV for 30 mM KCl, which means the membrane potential *V*_r_ of the reference solution (30 mM KCl + 0.3 mM tartaric acid) is higher than that of the 30 mM KCl sample solution. This is acceptable because the reference solution contains tartaric acid and is weakly acidic (pH about 3.5), which leads to a decrease in the dissociation of $H^{+}$ from PADE and an increase in the surface potential of the membrane.

The slope for KCl and selectivity coefficients of the sensors using blank membrane and PADE membranes based on 0.03 mM and 30 mM are summarized in Table 4. First, the response slope for KCl largely increases with increasing PADE concentration. It is quite reasonable [32,33] that the increase in PADE concentration can increase the charge density inside the membrane, which is closer to the ideal-type response. Next, the selectivity coefficients of the blank membrane and the low-PADE membrane electrodes are almost the same. Last, the high-PADE membrane electrodes exhibit low selectivity for different alkali metal ions, as can be seen from the very small differences in selectivity coefficients. These facts are detailed later in Section 3.3 in relation to the membrane impedance.

### 3.2. Sensor Response to Different Alkali Metals in the TFPB Membrane

Figure 6a,b show the sensor responses of the membrane electrodes with low (0.03 mM) and high (30 mM) TFPB concentrations to different alkali metal salts, respectively. The reference potentials (*V*_r_) of the measurements are shown in Table 5 and can be used to calculate the response value (*V*_s_ − *V*_r_).

First, both the responses of membrane electrodes increased logarithmically with increasing concentrations of each alkali metal salt. The slope for KCl shown in Table 5 is 51.4 mV in 30 mM TFPB and is 14.9 mV in 0.03 mM TFPB. The response of the 30 mM TFPB membrane is much higher than that of 30 mM PADE membrane, indicating that the charge density of the TFPB membrane is higher than that of PADE membrane [32,33], which is caused by the fact that TFPB is completely dissociated while the PADE is partially dissociated.

Next, Figure 6b shows that the response to 30 mM KCl is about 0 mV, which means that the membrane potential *V*_r_ of the reference solution (30 mM KCl + 0.3 mM tartaric acid) is the same as that of 30 mM KCl sample solution. It is quite reasonable because the dissociation of TFPB is complete and is not affected by pH.

Figure 7 shows the sensor response of the membrane electrodes with high PADE and TFPB concentrations (30 mM) to 100 mM NaCl samples at different pH levels. As the pH decreased, the response value of the PADE membrane increased, while the TFPB membrane remained unchanged. This can directly indicate the difference in dissociation of the two membranes as previously discussed.

Last, by comparing the response of the low-TFPB membrane (Figure 6a) with that of the blank membrane (Figure 5a), the low-TFPB membrane shows the same value around −100 mV at 0.1 mM salts as the blank one, but its responses to higher salt concentration do not exactly converge to those of the blank. In fact, lower concentrations such as 0.01 mM and 0.001 mM of TFPB membranes have also been tested experimentally, and the results were almost the same as those of 0.03 mM of TFPB membrane, indicating the TFPB membrane keeps its property even at the very low TFPB concentrations.

Table 5 shows the response slope for KCl and selectivity coefficients of the sensor using TFPB membrane based on low (0.03 mM) and high (30 mM) concentrations. The sensor of 0.03 mM TFPB has a low selectivity for alkali metal ions. However, the sensor of high TFPB concentration (30 mM) shows high selectivity for alkali metal ions and follows the Hofmeister series in the order ${\mathrm{Cs}}^{+}>K^{+}>{\mathrm{Na}}^{+}>{\mathrm{Li}}^{+}$, as is also shown in Figure 5b.

### 3.3. Selectivity Coefficient and Impedance of Different Concentrations of PADE Membrane

Figure 8 shows the selectivity coefficient ($k_{K,j}^{pot}$) as a function of the PADE concentration. It can be seen that the selectivity coefficient does not change much with increasing PADE concentration and always remains around unity. Additionally, according to Section 3.1, the response properties of PADE membranes at low PADE concentrations tend to be similar to the blank membrane and depend on NPOE; the membranes exhibit low selectivity but still can distinguish ${\mathrm{Cs}}^{+}$ from other alkali metal ions. At concentrations above 20 mM, however, the properties of the membranes are more biased towards PADE itself, which exhibits much lower ion selectivity for these four alkali metal ions.

Figure 9 shows the impedance of the PADE membrane with different concentrations. The impedance decreases with increasing lipid concentration, but remained almost unchanged beyond 3 mM. The magnitude of the impedance of PADE membranes is always maintained above $10^{6}\;\Omega \cdot {\mathrm{cm}}^{2}$. As the surface charge density increases with higher lipid concentrations, the electrical conductivity of the membrane increases, and thus the impedance decreases, as expected. However, lipid molecules PADE have the property of partial dissociation of $H^{+}$. Therefore, the $H^{+}$ dissociation was inhibited when there are too many lipids on the surface to reduce the mutual repulsion of homogeneous charges. As the impedance is related to the surface charge density, even if the PADE concentration increased, the impedance did not change anymore.

### 3.4. Selectivity Coefficient and Impedance of Different Concentrations of TFPB Membrane

Figure 10 shows the selectivity coefficient ($k_{K,j}^{pot}$) as a function of the TFPB concentration. It can be seen that when the concentration of TFPB is below 0.1 mM, the membranes hardly show ion selectivity. Beyond 0.1 mM TFPB, on the other hand, the membranes start to exhibit ion selectivity, which increases, reaching a maximum at a concentration of 3 mM and remaining essentially constant. As mentioned in Section 3.2, the ion selectivity of TFPB membranes follows the Hofmeister series in the order ${\mathrm{Cs}}^{+}>K^{+}>{\mathrm{Na}}^{+}>{\mathrm{Li}}^{+}$.

Many studies have shown that most liquid membrane electrodes such as ion-selective electrodes are ion-selective and follow the Hofmeister series [34,35,36,37,38]. The ion selectivity is affected by various factors such as valence, solvated equivalent volume, polarizability, complex formation of counterions in the aqueous solution. The experimental result in the negatively-charged monolayer [38] is typical because counterions largely affecting the surface potential compress the molecular surface area more effectively. Among these factors affecting the Hofmeister series, the hydration of ions, i.e., hydration energy is most important [38,39]. The order of ionic radius is ${\mathrm{Cs}}^{+}>K^{+}>{\mathrm{Na}}^{+}>{\mathrm{Li}}^{+}$, which is opposite to the order of hydrated radius; ${\mathrm{Li}}^{+}>{\mathrm{Na}}^{+}>K^{+}>{\mathrm{Cs}}^{+}$. The larger the hydrated radius of the ion, the stronger the binding force with water molecules (hydration energy). This larger binding force makes it more difficult for cations to interact with negatively-charged lipid molecules on the membrane surface, thus making the sensor response lower and causing ion selectivity, as found in Figure 10.

The ion selectivity is remarkable at moderate or high TFPB concentrations in Figure 10 because the increase in charged sites effectively interacting with counter cations. On the other hand, the ion selectivity kept low even at high PADE concentrations (Figure 8); it is due to inhibition of $H^{+}$ dissociation of concentrated PADE molecules, to inhibit increasing electric repulsion between PADE molecules, resulting in effective charged sites not increasing as much. In contrast, the ion selectivity increased in the TFPB membrane, because TFPB molecules make complete ionization, as above.

Figure 11 shows the impedance of the TFPB membrane with different concentrations of TFPB. As the TFPB concentration increased, the membrane impedance largely decreased by three digits. With TFPB lower than 0.3 mM, the impedance remained above $10^{6}\;\Omega \cdot {\mathrm{cm}}^{2}$; however, it was below $10^{5}\;\Omega \cdot {\mathrm{cm}}^{2}$ when TFPB was higher than 3 mM. This is because the increasing charge density at the membrane surface with increasing concentration of completely-dissociated TFPB resulted in an increased electrical conductivity of the membrane. A rebound in impedance occurred when the TFPB concentration reached 300 mM. The reason for the rebounding impedance is considered to be the precipitation of TFPB, as an oil film was observed on the membrane surface, causing a reduction of electrical conductivity.

The membranes with impedances as high as $10^{6}\;\Omega \cdot {\mathrm{cm}}^{2}$ showed low ion selectivity, while those with low impedances below $10^{5\;}\Omega \cdot {\mathrm{cm}}^{2}$ exhibited high ion selectivity following the Hofmeister series. This property of the TFPB membranes with high impedances is the same as that of the PADE membranes showing high impedances at any PADE concentration.

### 3.5. The Advantage of Using TFPB in the Taste Sensing System

Based on what has been discussed in previous sections, TFPB can behave in different sensing patterns with its concentration being specifically controlled. The benefits of using TFPB as a component of the lipid polymer membrane have also been revealed.

Figure 12 shows the performance of the taste sensing system using membranes composed of TFPB in comparison to other membranes developed before. Figure 12a shows the responses of the TFPB membrane and commercialized CMTE membrane to a typical bitter substance quinine in distilled water. The CMTE membrane is a kind of ion-exchange membrane and is widely used in sewage disposal. The responses at all concentration steps tended to be flat, while the TFPB membrane showed great responses when the concentration of quinine exceeds 0.1 mM, which is considered to be a decent candidate of bitterness sensing membranes. Figure 12b shows the responses of membranes using TFPB and PADE to quinine in standard solution. It can be found that PADE is a better candidate for a bitterness sensing membrane, with the larger response and smoother linear shape for under 1 mM quinine, and the TFPB membrane started to behave effectively beyond 0.1 mM, making it a better choice when the concentration of the solution is large.

## 4. Conclusions

In this study, we investigated the differences in electrical properties of a membrane based on partially dissociated PADE and a membrane based on completely dissociated TFPB. The PADE membranes consistently exhibited low ion selectivity, irrespective of the PADE concentration, whereas the TFPB membranes exhibited low ion selectivity below 0.3 mM TFPB, and then showed significantly high ion selectivity above 1 mM and reached a maximum at 3 mM. The sequence of ion selectivity of the TFPB membrane is ${\mathrm{Cs}}^{+}>K^{+}>{\mathrm{Na}}^{+}>{\mathrm{Li}}^{+}$, which is the same as the Hofmeister series. It can be considered that the hydration of ions affects the electrostatic interaction with lipid molecules on the membrane surface.

In addition, the impedance of PADE membranes decreased from $3\times 10^{6}\;\Omega \cdot {\mathrm{cm}}^{2}$ to $1\times 10^{6}\;\Omega \cdot {\mathrm{cm}}^{2}$ and remained unchanged beyond 3 mM PADE, while the TFPB membranes continued to largely decrease from $3\times 10^{6}\;\Omega \cdot {\mathrm{cm}}^{2}$ to $3\times 10^{3}\;\Omega \cdot {\mathrm{cm}}^{2}$. This different property is caused by the difference in the dissociation state of the two membranes.

PADE molecules are used in the membranes of taste sensors for sourness, bitterness, and umami [11,12]. It was already clarified [21] that the response of the membrane using PADE to bitter substances deteriorated due to the acidic environment produced by $H^{+}$ dissociation of PADE, and this problem was overcome by developing the bitterness membrane using TFPB. Utilization of completely-dissociated materials such as TFPB will be also useful to control the sensitivity to taste substances, as discussed [21,40]. Furthermore, taste sensors should respond to taste itself in the coexistence of various ion species; hence, those with low hydration energy can possibly interfere with the detection of the taste produced by other substances. Therefore, adjustment of the concentration of membrane components and use of other lipid species with different ionization states will be effective for attaining the purpose to quantify the taste selectively. The results obtained here will contribute to the development of novel receptive membranes of taste sensors, the properties of which do not change based on some real conditions such as the use time or pH.

## Figures and Tables

**Figure 1 sensors-21-08343-f001:**
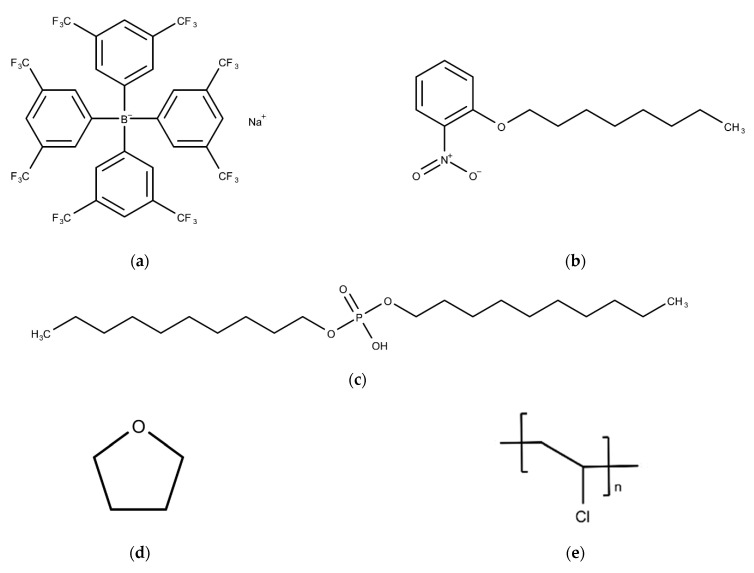
Structure of membrane components: (**a**) TFPB; (**b**) NPOE; (**c**) PADE; (**d**) THF; (**e**) PVC.

**Figure 2 sensors-21-08343-f002:**
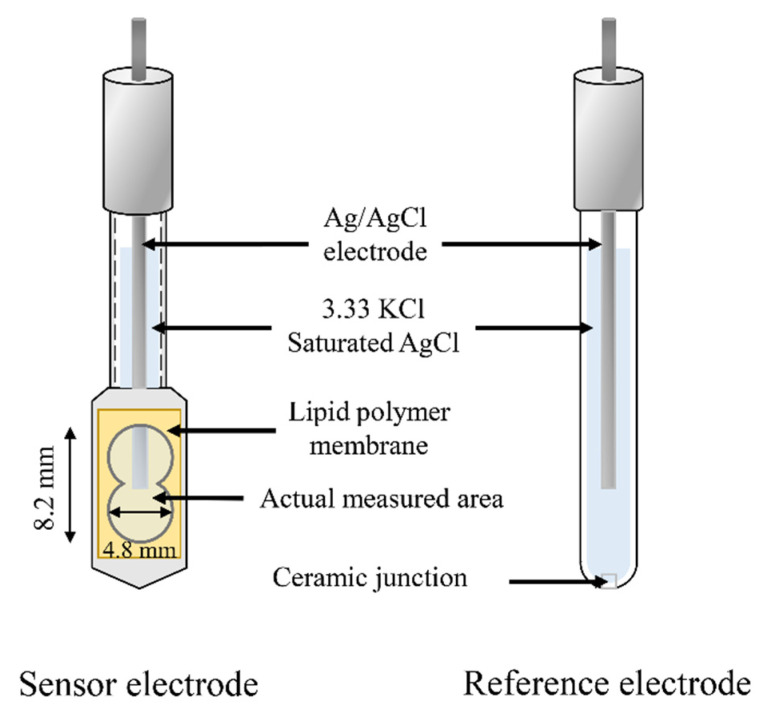
Constructions of taste sensor electrodes.

**Figure 3 sensors-21-08343-f003:**
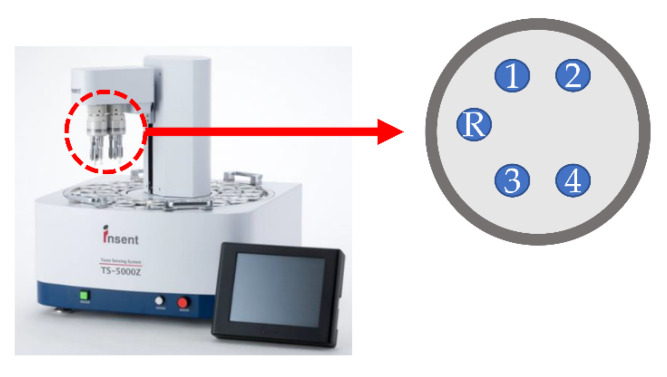
Taste sensor system TS-5000Z. One measurement cell could contain at most 4 sensor electrodes and 1 reference electrode. The 4 sensor electrodes do not need to use exactly the same membrane. In this study, the membranes used in the 4 sensor electrodes of one measurement cell were the same in measuring the samples and calculating the standard deviations.

**Figure 4 sensors-21-08343-f004:**
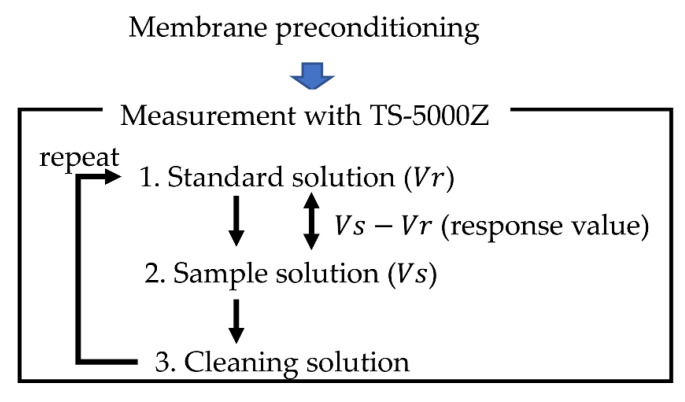
Measurement procedure of taste sensor.

**Figure 5 sensors-21-08343-f005:**
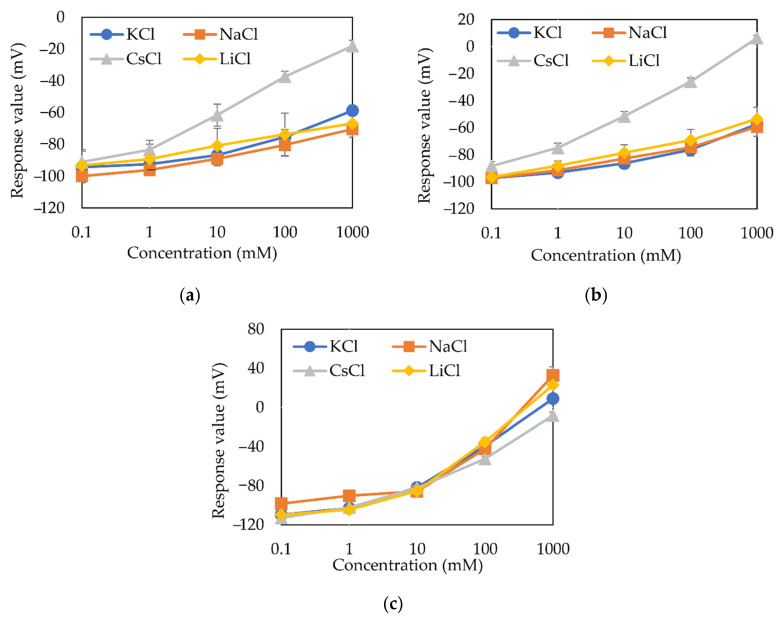
Sensor responses (*V*_s_ − *V*_r_) to alkali metal salts samples: (**a**) Blank membrane; (**b**) low PADE concentration (0.03 mM); (**c**) high PADE concentration (30 mM).

**Figure 6 sensors-21-08343-f006:**
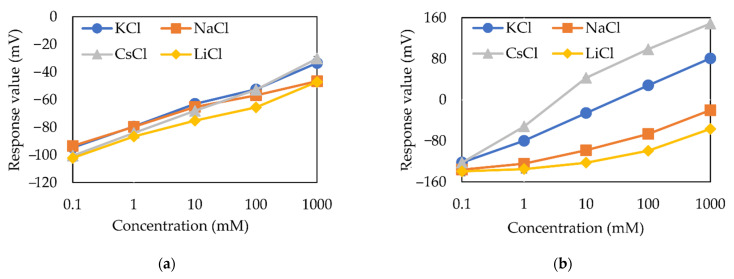
Sensor responses (*V*_s_ − *V*_r_) to alkali metal salts samples: (**a**) Low TFPB concentration (0.03 mM); (**b**) high TFPB concentration (30 mM).

**Figure 7 sensors-21-08343-f007:**
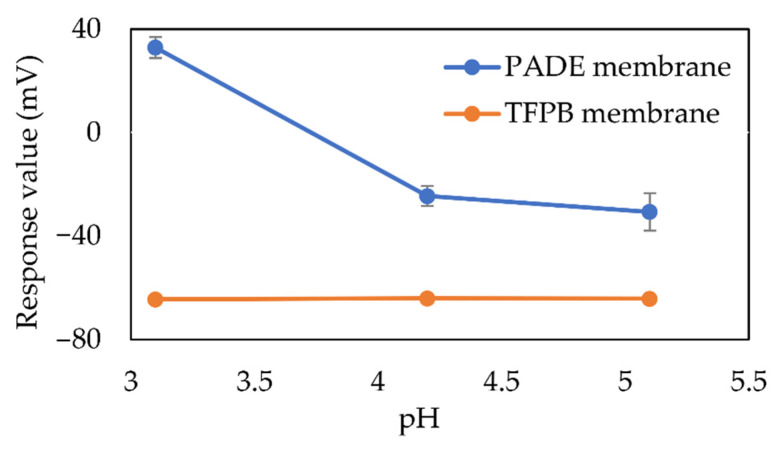
Sensor responses (*V*_s_ − *V*_r_) to 100 mM NaCl sample at different pH.

**Figure 8 sensors-21-08343-f008:**
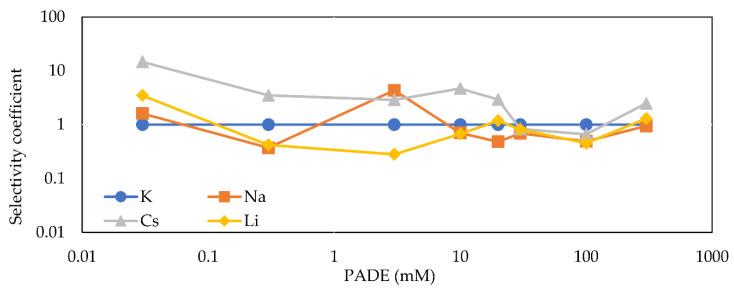
Variation of the selectivity coefficient ($k_{K,j}^{pot}$) with different PADE concentrations.

**Figure 9 sensors-21-08343-f009:**
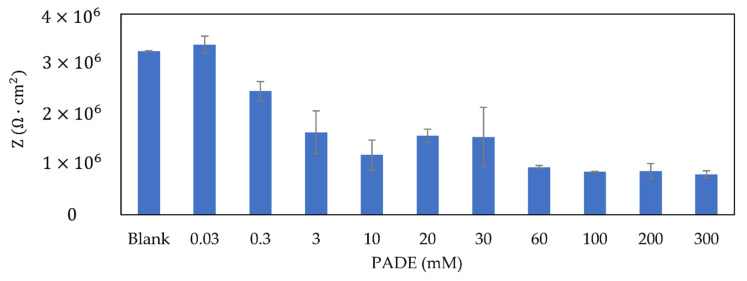
The impedance of the PADE membrane with different PADE concentrations.

**Figure 10 sensors-21-08343-f010:**
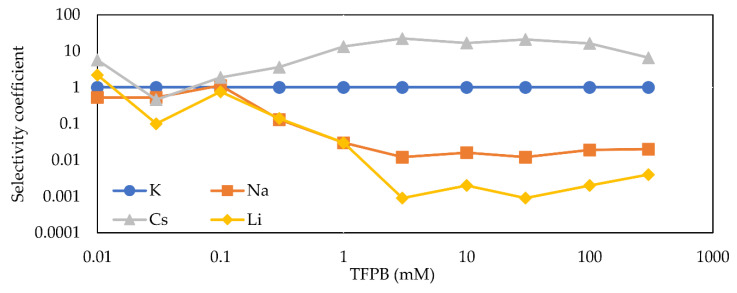
Variation of the selectivity coefficient ($k_{K,j}^{pot}$) with different TFPB concentrations.

**Figure 11 sensors-21-08343-f011:**
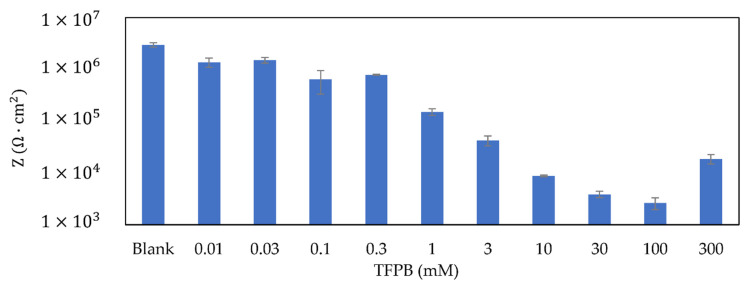
The impedance of the TFPB membrane with different TFPB concentrations.

**Figure 12 sensors-21-08343-f012:**
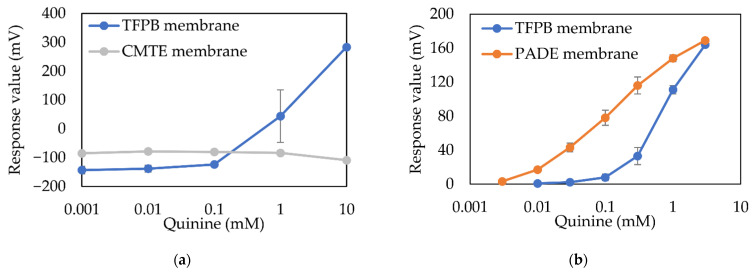
The examples using TFPB membranes and other membranes to measure bitterness samples: (**a**) TFPB (30 mM) membrane and CMTE membrane; (**b**) TFPB (15 mM) membrane and PADE (35 mM) membrane, respectively.

**Table 1 sensors-21-08343-t001:** Concentrations of TFPB and PADE in membranes.

Membrane	Concentration
Blank membrane	0 (NPOE + PVC)
PADE membrane	0.03, 0.3, 3, 10, 20, 30, 60, 100, 200, 300 mM
TFPB membrane	0.01, 0.03, 0.1, 0.3, 1, 3, 10, 30, 100, 300 mM

**Table 2 sensors-21-08343-t002:** Concentrations of alkali metal salts samples (solvent: pure water).

Sample	Concentration
KCl	0.1, 1, 10, 100, 1000 mM
NaCl	0.1, 1, 10, 100, 1000 mM
LiCl	0.1, 1, 10, 100, 1000 mM
CsCl	0.1, 1, 10, 100, 1000 mM

**Table 3 sensors-21-08343-t003:** Measurement details and parameters of membrane impedance.

Details	Parameter
Measure procedure	NOVA FRA impedance potentiostatic
Electrode configuration	Three electrode system
Electrolyte	30 mM KCl + 0.3 mM tartaric acid
Sweeping frequencies	0.1 Hz, 0.223 Hz, 0.5 Hz
Voltage	200 mV

**Table 4 sensors-21-08343-t004:** The slope for KCl, reference potential (*V*_r_), and selectivity coefficients of the sensors.

Membrane Component	Blank Membrane	PADE 0.03 mM	PADE 30 mM
Slope for KCl (mV)	9.0	9.7	30.1
*V*_r_ (mV)	28.5	25.7	−29.9
**Selectivity coefficients (** $logk_{K,j}^{pot})$			
$K^{+}$	0	0	0
${\mathrm{Na}}^{+}$	−0.3	0.2	−0.2
${\mathrm{Li}}^{+}$	0.5	0.5	−0.1
${\mathrm{Cs}}^{+}$	1.2	1.2	−0.1

**Table 5 sensors-21-08343-t005:** The slope for KCl, reference potential (*V*_r_), and selectivity coefficients of the sensors.

Membrane Component	TFPB 0.03 mM	TFPB 30 mM
Slope for KCl (mV)	14.9	51.4
*V*_r_ (mV)	78.5	−109.9
**Selectivity coefficients (** $logk_{K,j}^{pot})$		
$K^{+}$	0	0
${\mathrm{Na}}^{+}$	−0.3	−1.8
${\mathrm{Li}}^{+}$	−1	−2.7
${\mathrm{Cs}}^{+}$	−0.3	1.2

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
