# Peer review of "Electrical Properties of Two Types of Membrane Component Used in Taste Sensors"

_sensors, 2021, doi:10.3390/s21248343_

Round 1

Reviewer 1 Report

This manuscript presents comparison of electrical properties of PADE and TFPB membranes used in taste sensor system. The experiments and methodologies were explained in detail. The authors studied and compared the voltage response, impedance response, selectivity of the two membranes used for taste sensors. The idea is simple and seems effective. This manuscript could be suitable for this journal.

However, there is a major issue that the manuscript only compares the electrical properties of the two membranes used for taste sensing. It is inadequate to just present the properties of the component in the sensor system. I would suggest adding the result of the sensor system, which will fit the scope of the journal better. There should be one result at the end to show TFPB is good or better to be applied in a taste sensor system, which will make this a complete story.

Author Response

Dear referee,

Thank you for your suggestion and comments. Based on your suggestion, we have added Section 3.5 in the article about the advantage of using TFPB in the taste sensing system. You can see it in the attached file.

I hope this will complete the story.

Best regards,

Authors

Author Response

Dear referee,

Thank you for your suggestion and comments. We have revised the article based on your suggestion. And please allow me to answer your question here.

Best regards,

Authors.

1- Please consider including the raw data of the measurements.

A: We have added the reference potential (Vr) in Table 4 and Table 5. Because the date of the figures is response value (Vs-Vr), the membrane potential (Vs) can be calculated.

2- It would improve the discussion if real samples were tested with the optimal concentration from both membranes

A: We have added Section 3.5 in the article about using TFPB in the taste sensing system. 

3- Please clarify how the membrane is fixed on the sensor probe.

A: We added Line 101 through Line 103 to explain this.

4- “To investigate the effect of lipid concentration on the electrical properties of membranes, the PADE membranes were fabricated with ten different concentrations from 0.03 mM to 300 mM, and the TFPB membranes from 0.01 mM to 300 mM.” please explain why different concentrations were used.

A: We added Line 93 through Line 97 to explain this.

5- “immersed in standard solution (30 mM KCl and 0.3 mM tartaric acid) for 30 s to obtain a membrane potential called Vr. In step 2, the measurement cell was immersed in sample solution for 30 s to obtain a new membrane potential called Vs. The difference between Vs and Vr (Vs-Vr) is defined as the relative value.” 

a. why the reference measurement had different pH from the samples? 

A: The standard solution contains a small amount of salt (30 mM KCl) to stabilize the sensor and a small amount of acid (0.3 mM tartaric acid) to stabilize the pH of the standard solution itself.

b. How to ensure the pH difference is not hindering the signal? 

A: During the entire measurement process, the reference potential was always maintained the same before and after measuring the sample

c. Please present at least one measurement without tartaric acid (full calibration curve, vs-vr, for at least one type of cation) 

A: We did not do this experiment because the lack of tartaric acid would affect the stability of the measurement.

d. Why the measurement time was so short? 

e. If the sensor needs such a long time to stabilize during preconditioning, why such a small measurement interval was used?

A: We think the measurement time of 30 s is enough because the surface of the membrane (such as lipid) has already been stabilized through the pre-conditioning.

f. how to analyze drifting (with only 30sec data, not shown)?

A: We think the membranes do not drift.

6- “In step 3, to restore the electrode surface to its pre-measurement condition, the measurement cell was washed with a cleaning solution (100 mM HCl and 30 vol% ethanol).” 

a. Please explain the cleaning process, what is the role of HCl? 

A: The taste sensor in this paper used negatively charged lipid membranes. In these negatively charged lipid membranes, the adsorbed substances are divalent positive ions (calcium and magnesium ions) and positively charged bitter substances. On the one hand, Divalent calcium and magnesium ions are thought to bind two POO groups, and HCl replaces its H+ to make the POO group become POOH. On the other hand, the positively charged bitter substances are electrically attracted to the negatively charged lipid membrane, and the hydrophobic part of the lipid membrane is hydrophobically bound to the hydrophobic part of the bitter substances. Hydrophobically bound bitter substances are easily dissolved by ethanol. Thus we used HCl and ethanol to clean the negatively charged lipid membranes.

b. Why does the sensor need to be cleaned? 

A: The target of measurement in this paper is monovalent ions, and in this case, cleaning is not necessary. However, the measurements in this paper followed the normal measurement procedure for general foods. General foods contain divalent calcium and magnesium ions and bitter substances. Divalent calcium and magnesium ions take a long time to remove, so they need to be replaced by acid as described above. In addition, bitter substances are adsorbed to the lipid membrane through hydrophobic bonds, so it takes several hours or more for the bitter substances to be removed from the membrane if there is no cleaning process. Therefore, cleaning is necessary.

c. After cleaning, was new preconditioning performed? Any preconditioning between cycles?

A: No there was not. The cleaning process did not affect the membrane surface. This can be confirmed from the fact that the potential of the standard solution measured after cleaning was the same as before.

7- “measurement cell contains 4 sensor electrodes and 1 reference electrode, and five measurement cycles are needed for one measurement procedure. The average of the third to fifth measurement” what are the 4-sensor electrode?

A: We added Figure 3 in the article to explain this.

8- “pure water, which means the pH of the sample solutions is about 5.2. The selectivity coefficients were calculated using the separate solution method (SSM) [26], where the response values from 10 mM to 1000 mM were utilized and those of potassium chloride were adopted as the reference.” Why was a different reference solution used for selectivity? No tartaric acid?

A: the reference potential (Vr) was the reference at the time of measurement to calculate the response value (Vs-Vr), while the selectivity is calculated from the response curves (Figure 5 and Figure 6) of the four cations after measurements. The potassium chloride’s response curve was chosen as the reference to calculate the selectivity because it had the most stable response (straight). The reason for using 10 mM to 1000 mM response for each ion in the selectivity calculation is to exclude the increase in resistance of the solution when the ion concentration is too small, which makes the response unstable.

9- “0.3 mM tartaric acid) is higher than that of 30 mM KCl sample solution. This is acceptable because the reference solution contains tartaric acid and is weakly acidic (pH about 3.5), which leads to a decrease in the dissociation of H+ from PADE and an increase in the surface potential of the membrane”

a. Please include a control without tartaric acid to confirm this claim. 

A: Sorry we did not do this experiment.

b. If the dissociation is decreased, the polymer membrane should be more neutral? how the potential increase with decreased dissociation? 

A: The taste sensor measures and outputs the surface potential of a lipid polymer membrane. When the membrane surface is negatively charged due to lipid dissociation, and dissociation is inhibited (dissociation decreased), the surface potential shifts to 0 mV (potential increased).

c. Why solutions cannot be measured consecutively without the need for a reference?

A: It can be used for continuous measurement, but to improve the measurement accuracy, it is calibrated at one point with a standard solution each time a sample is measured.

d. Does the measurement protocol is compatible with the actual application of the sensor?

A: Yes.

10- “the slope for KCl and selectivity coefficients of the sensors using blank membrane, PADE membranes based on 0.03 mM and 30 mM are summarized in Table 4.” Were the measurements made using Vs-Vr?

A: Yes.

11- “It is quite reasonable because the dissociation of TFPB is complete and is not affected by pH” Please consider including a calibration curve for cations in different pHs and compare the response for both membranes. This result would enhance the discussion and make the pH dependence comparison stronger. 

A: We added Figure 7 and Line 195 through Line 198 to enhance the discussion.

12- “On the other hand, the ion selectivity kept low even at high PADE concentrations (Figure 6); it is due to inhibition of dissociation of concentrated PADE molecules, resulting in effective charged sites not to increase.” is this sentence contrasting with the previous discussion?: “This is acceptable because the reference solution contains tartaric acid and is weakly acidic (pH about 3.5), which leads to a decrease in the dissociation of H+ from PADE and an increase in the surface potential of the membrane.” Please discuss. 

A: We changed the sentence “it is due to inhibition of dissociation of concentrated PADE molecules, resulting in effective charged sites not to increase.” to “it is due to inhibition of   dissociation of concentrated PADE molecules, to inhibit increasing electric repulsion between PADE molecules, resulting in effective charged sites not to increase so much.”

13- Please consider revising the references for more up to date publication

A: We added some up-to-date references. 

Round 2

Reviewer 1 Report

The authors have addressed all my concerns. I would recommend accepting in present form of this manuscript.